# Open and minimally invasive surgery for gastrointestinal stromal tumours: a systematic review and network meta-analysis protocol

Mingchun Mu ,[1] Zhaolun Cai ,[1] Chunyu Liu,[2] Chaoyong Shen,[1] Yuan Yin,[1] Xiaonan Yin,[1] Zhiyuan Jiang ,[1] Zhou Zhao,[1] Bo Zhang [1,3]

MM, ZC and CL contributed equally.

¹Department of Gastrointestinal Surgery, Sichuan University West China Hospital, Chengdu, Sichuan, China
²Department of Pharmacy, Sichuan University West China Second University Hospital, Chengdu, Sichuan, China
³Sanya People's Hospital/West China (Sanya) Hospital, Sichuan University West China Hospital, Sanya, Haina, China

**Correspondence to**
Professor Bo Zhang;
hxwcwk@126.com

## ABSTRACT

**Introduction** Gastrointestinal stromal tumours (GISTs) are the most common mesenchymal tumours of the digestive system, and complete resection is the only way to provide a radical cure for resectable GISTs. Open surgery and minimally invasive approaches, including laparoscopy, robotic surgery and endoscopy, consist of the mainstream GIST resection. However, there is still a lack of evidence regarding which surgical outcomes and long-term prognosis would be better. Thus, we are planning to conduct a network meta-analysis and systematic review aiming to determine the comparative effectiveness among laparotomy, laparoscopy, endoscopy, robotic surgery, and laparoscopic and endoscopic cooperative surgery in GISTs.

**Method and analysis** PubMed, EMBASE, the Cochrane Library and Web of Science will be searched for published studies to identify the proper literature comparing open resection, laparoscopy, endoscopy, robotic surgery, and laparoscopic and endoscopic cooperative surgery for resecting GISTs from inception to February 2021. Randomised controlled trials (RCTs) and non-randomised studies comparing at least two different interventions for GIST resection will be included. RCTs and non-randomised studies will be synthesised and analysed separately. Bayesian network meta-analysis will be performed to compare the surgical outcomes and long-term prognosis among the resection methods above. The included studies will be divided into several subgroups according to tumour location and size for further analysis. Sensitivity analysis will be performed to identify and explain heterogeneity to make our results robust. Meta-regression will serve as a supplementary method if data are available. The quality of evidence will be evaluated by the Grading of Recommendations, Assessment, Development and Evaluation.

**Ethics and dissemination** No ethical approval is required for this network meta-analysis, as it is based on already published data. The findings of the review will be published in a peer-reviewed journal.

**PROSPERO registration number** CRD42021237892.

### Strengths and limitations of this study

► This is the first Bayesian network meta-analysis and systematic review comparing the efficacies of different types of surgical approaches for resection of gastrointestinal stromal tumours.
► Randomised controlled trials and non-randomised studies will be included and analysed separately to strengthen the statistical power.
► The Grading of Recommendations Assessment, Development and Evaluation approach will be used to evaluate the quality of evidence to provide comprehensive suggestions and references for clinical decision making and guideline development.
► Our results will be limited by the quantity and quality of eligible studies included.

approximately 10–15 people per 1 million.[1] Targeted therapy with tyrosine kinase inhibitors based on gene mutation and risk classification has been shown to be effective in prolonging life and delaying recurrence or metastasis.[2–4] However, for GISTs, once tumours are estimated to be resectable, complete resection is the only way to achieve a radical cure.[5]

For GISTs, open resection, which is the typical surgery used to remove the tumour, has historically been the primary method used due to its clear surgical field and feasibility. Furthermore, as general surgery has trended towards minimally invasive surgery, many minimally invasive approaches, including laparoscopy, endoscopy and robotic surgery, have increased in popularity. Laparoscopy, which has been demonstrated to have a lower incidence of perioperative events and indistinctive long-term complications than traditional surgery, has become the main trend.[6] Serendipitously, because asymptomatic GISTs are located in feasible sites, resection by endoscopy has also gained acceptance, as this

## INTRODUCTION

Gastrointestinal stromal tumours (GISTs) are the most common mesenchymal tumours of the digestive system, with a prevalence of

method is even less invasive; however, it has been reported that the incidence of positive margins under endoscopy is still a problem that needs to be solved.[7] Furthermore, laparoscopic and endoscopic cooperative surgery would also be a choice for surgery by experienced physicians if the tumour is located in a specific site.[8] Additionally, robotic surgery is an option due to its prominent view and remarkable coordination, although as a relatively new operation, it has a lengthy learning curve.[9 10]

These approaches have been widely used for GIST resection. The selection of the operation type is determined by tumour sites, tumour size, and surgeon preference, and there is no consensus about the preferred approaches for different locations and magnitudes.[11] Although there are considerable traditional pairwise meta-analyses discussing two of them, most of these studies are limited to open vs laparoscopic surgery or laparoscopy versus endoscopy.[12–16] In other words, regrettably, there is still a lack of evidence regarding which surgical outcomes and long-term prognosis will be better than those of open resection, laparoscopy, endoscopy, robotic surgery and laparoscopic and endoscopic cooperative surgery for GISTs at different sites and with different tumour sizes.

Thus, we are planning to conduct this network meta-analysis and systematic review with the aim to synthesise all the evidence available to enlarge the sample size and identify the best strategy among the five types of resection mentioned above for GISTs. Additionally, in contrast to side-to-side pairwise meta-analysis, network meta-analysis could differentiate three or more methods by not only direct but also indirect comparison, which could obtain the utmost use of existing publications. Moreover, to further guide clinical practice, in our study, several subgroups will be generated according to tumour location and size to discuss safety and efficiency. To further guide clinical practice, all the outcomes supported by this network and systematic review will be evaluated by the Grading of Recommendations, Assessment, Development and Evaluation (GRADE) tool to rank different treatments.[17]

## METHODS AND ANALYSIS
### Design
The protocol of network meta-analysis is guided by the Preferred Reporting Items for Systematic Reviews and Meta-Analyses Protocols.[18] Network meta-analysis and systematic review will be conducted using Bayesian network meta-analysis.

### Information resources and search strategy
The following databases will be searched for published studies: PubMed, EMBASE, and the Cochrane Library from inception to February 2021 without language restrictions. Medical subject headings terms (Mesh) combined with text words and synonyms will be performed in our search course. In addition, the manual search and reference search will be performed to enlarge the search range. A draft search strategy for PubMed is presented in online supplemental material 1.

### Eligibility criteria/exclusion criteria
The inclusion criteria based on the patients, intervention, comparison, outcomes, and study design framework are as follows:

### Participants
The study will include adult patients (≥18 years) with a diagnosis of GISTs according to pathology.

### Interventions/comparators
This study will include studies comparing at least two different interventions among the following interventions: open resection, laparoscopy, endoscopy, robotic surgery and laparoscopic and endoscopic cooperative surgery to resect GISTs. Endoscopic resection is defined as any resection under endoscopy, such as endoscopic submucosal dissection, endoscopic full-thickness resection, submucosal tunnelling endoscopic resection and other types of resection only via endoscopy.

### Outcomes
The primary outcomes will be disease free survival （DFS）, positive margin rate and tumour rupture.
The second outcome is as follows:
1. Surgical outcome: primary surgical outcomes are procedure time and surgical blood loss; the second surgical endpoint was that whether or not there was conversion to another resection.
2. Postoperative outcomes: postoperative complications (per Clavien-Dindo grade), hospital stay, time to flatus, time to liquid and time to soft diet.
3. Survival: recurrence rate and overall survival (OS).

The time point for outcomes will be the longest follow-up time in each study.

### Study designs
This study will include non-randomised studies (NRSs), including prospective or retrospective cohort studies and randomised controlled trials (RCTs). RCT and NRS studies will be synthesised and analysed separately. We will include full-text publications, results published in non-commercial trial registries and abstracts if sufficient information is available on study design, characteristics of participants, interventions and outcomes. We will contact study investigators to request missing data.

Exclusion criteria are as follows:
1. Reviews, comments, letters and animal studies.
2. Studies from the same institution or with overlapping patients. In such cases, we would review the including criteria and study period to make sure whether a single patient was overlapped. If yes, efforts will be made to contact the authors to get answers. Otherwise, only the latest or the most comprehensive one will be included.

## Study selection and data extraction

Two authors (MM and CL) will independently screen the titles and abstracts to assess the eligibility of all studies. Questionable articles will be subject to a full-text review to gain more information. Disagreement will be resolved by a third assessor (ZJ) until consensus is reached among the three authors. Only studies meeting the eligibility criteria will be finally included.

The following information will be extracted using a standard form: first author, publication year, study design, number of patients, tumour site, tumour size (cm), mitotic index (/50 high-power field), risk classification; resection approaches, resection range, conversion rate, operation time (min), blood loss (mL), length of hospital stay (days), time to flatus (days), time to liquid (days), time to soft diet (days); number and rate of perioperative complications; number and rate of patients with positive margins; follow-up time (months); number and rate of patients with recurrence; and OS. Details regarding consultation with a third author until consensus is reached among three authors or contact with the original authors for further information will be documented. To ensure reproducibility, the reasons for the removal of any study after a full-text review will be recorded in online supplemental document.

## Risk of bias for included studies

1. For NRSs

   The tool of risk of bias in non-randomised studies of interventions (ROBINS-I) will be used to estimate the risk of bias of the included prospective or retrospective cohort studies.[19] Seven domains of bias throughout the entire course of intervention were well evaluated in this tool: (1) bias due to confounding, (2) bias in selection of participants into the study, (3) bias in classification of interventions, (4) bias due to deviations from intended intervention, (5) bias due to missing data, (6) bias in measurement of outcomes and (7) bias in selection of the reported result. Overall bias after seven domains will be estimated. On the condition of comprehensive consideration above, each individual included study will be assessed as having the low, moderate, serious and critical risk of bias. If critical information is lacking for the evaluation of the risk of bias, such studies will be estimated as having no information.

2. For randomised studies

   The risk-of-bias tool from Cochrane Handbook V.5.1.0 will also be used if random controlled trials are included. Six domains of risk of bias will be evaluated as follows: random sequence generation, allocation concealment, blinding, incomplete outcome data, selective reporting and other bias. Each eligible study with abundant information will be judged as having a low or high risk of bias. Otherwise, it will be evaluated as unclear.[20]

   The risk-of-bias assessment will be completed by two independent reviewers (MM and CL), and conflicts will be resolved by a third reviewer (ZJ) until consensus is reached among the three authors.

## Small sample effects

Comparison-adjusted funnel plots will be drawn to detect the small sample effects on the results.[21]

## Dealing with missing data

For missing data, attempts to obtain more information from original authors will be made. In the absence of a reply, we will try to calculate the data through the available coefficients according to the Cochrane Handbook for Systematic Reviews. For continuous outcomes, SDs will be estimated by stand errors, p values or CIs, depending on how the original research is provided. Otherwise, SDs will be evaluated based on the median or IQR.[22] The potential impact of these missing data will be tested by sensitivity analysis.

## Statistical analyses

When quantitative analysis cannot be conducted, we will narratively describe the results. If the quantitative analysis is feasible, subsequent statistical analyses will be conducted. RCTs and NRSs will be synthesised and analysed separately.[23]

1. Geometry of the network

   A network plot will be drawn to describe and present the geometry of types of interventions, including open resection, laparoscopy, endoscopy, robotic surgery and laparoscopic and endoscopic cooperative surgery.

2. Assessment of transitivity

   A narrative summary will be presented to describe the characteristics of each included study. To assess transitivity, we will compare the distributions of baseline participant characteristics across studies and treatments to confirm that they are parallel among different comparisons.

3. Direct comparison

   A traditional pairwise meta-analysis will be performed when at least two studies exist for an outcome by STATA V.12.0 software (STATA). The DerSimonian-Laird method and random effects model will be used.[24] The $\chi 2$ test and $I^2$ statistic will be applied to quantify the extent of between-trial heterogeneity. $I^2 > 50\%$ or $p < 0.1$ will indicate considerable heterogeneity.

4. Indirect and mixed comparison

   Network meta-analysis will be conducted using a Bayesian Markov chain Monte Carlo framework and fitted in R software with the gemtc package.[25 26] Dichotomous data will be determined by using OR with the 95% CrI. Continuous outcomes will be analysed using weighted mean differences or standardised mean differences if different measurement scales are used. Surface under the cumulative ranking area values will be used to rank the different resection methods.[27 28]

5. Assessment of inconsistency

   For closed-loop network meta-analysis, direct and indirect comparisons coexist; thus, it entails the assessment

of inconsistency to reflect differences between the two. In our study, the node-splitting method, which involves splitting mixed evidence into direct and indirect evidence in each node for comparison, will be used to assess inconsistency.[29 30] If a discrepancy is not found, this network meta-analysis can be considered to fit the consistency model. On the other hand, when a significant difference between direct and indirect evidence occurs, an inconsistency model will be used, and potential reasons for inconsistency will be discussed.

6. Subgroup analysis

Subgroup analysis will be used to identify and explain the source when significant heterogeneity is detected. Meta-regression serves as a further supplementary method if data are available. Preliminary subgroups are as follows:

– Tumour location (stomach, small intestine, colon and rectum).
– Tumour size (less than 5 cm and more than 5 cm).
– Surgical approaches (tumour resection only vs radical organ resection).
– Adjuvant treatment (yes or no).

7. Sensitivity analysis

Sensitivity analysis will be performed to check the stability by excluding studies with a high risk of bias if possible.

## Quality of evidence

The quality of evidence will be assessed by the GRADE tool for rating the quality of treatment effect estimations from the network meta-analysis.[17] Based on risk of bias, inconsistency, indirectness, imprecision and publication bias, the quality of evidence will be rated as high, moderate, low or very low.

## ETHICS AND DISSEMINATION

The network meta-analysis and systematic review are based on published data, so ethical approval is not a requirement.

Our findings will be published in a peer-reviewed journal. This network analysis and systematic review is now in progress; it will start on 19 February 2021, and the expected end time is 19 October 2021.

## Patient and public involvement statement

Patients or members of the public were not involved with the design of this study.

**Contributors** BZ and ZJ designed this study, and BZ is the guarantor for the article. MM, CL and ZZ drafted the protocol, the draft was modified by CS and YY. MM and CL will search, select, and identify studies included, and extract data independently, while XY will be the third reviewer for study selection and data extraction. BZ will serve as an adviser for methodology. All authors have approved the publication of this protocol.

**Funding** This study is funded by the Project Sichuan Science and Technology Support Programme (Grant No. 2020YFS0234, 2020YFS0233) and the 1.3.5 project for disciplines of excellence, West China Hospital, Sichuan University (ZYJC18034). This project is also supported by Hainan Province Clinical Medical Centre.

**Competing interests** None declared.

**Patient consent for publication** Not applicable.

**Provenance and peer review** Not commissioned; externally peer reviewed.

**ORCID iDs**
Mingchun Mu http://orcid.org/0000-0003-0133-8882
Zhaolun Cai http://orcid.org/0000-0002-3706-6703
Zhiyuan Jiang http://orcid.org/0000-0003-2483-3413
Bo Zhang http://orcid.org/0000-0002-0254-5843

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
