## [Reviewer comments · BMJ Open]

ARTICLE DETAILS

TITLE (PROVISIONAL)	Open and minimally invasive surgery for gastrointestinal stromal tumours: A systematic review and network meta-analysis protocol
AUTHORS	Mu, Mingchun; Cai, Zhaolun; Liu, Chunyu; Shen, Chaoyong; Yin, Yuan; Yin, Xiaonan; Jiang, Zhiyuan; Zhao, Zhou; Zhang, Bo

VERSION 1 – REVIEW

REVIEWER	Dudzisz-Śledź, Monika Maria Skłodowska-Curie National Research Institute of Oncology in Warsaw, Bone and Soft Tissue Sarcomas and Melanoma
REVIEW RETURNED	22-Apr-2021

GENERAL COMMENTS	The authors plan to conduct an interesting and meaningful analysis and review, especially in the era of developing new surgical technologies in recent years. All the topics mentioned in the review checklist seem to be adequately covered. Concerning statistical methods of analysis, this publication may require an additional review by a statistician. I don't see any significant concerns about the publication. Nevertheless, I have a few suggestions listed below. Please consider including some more available papers dedicated to comparison of open and laparoscopic surgery in GIST treatment into references, like: - Chen K, Zhou YC, Mou YP, Xu XW, Jin WW, Ajoodhea H. Systematic review and meta-analysis of safety and efficacy of laparoscopic resection for gastrointestinal stromal tumors of the stomach. Surg Endosc. 2015 Feb;29(2):355-67. doi: 10.1007/s00464-014-3676-6. Epub 2014 Jul 9. PMID: 25005014- Cui JX, Gao YH, Xi HQ, et al. Comparison between laparoscopic and open surgery for large gastrointestinal stromal tumors: A meta-analysis. World J Gastrointest Oncol. 2018;10(1):48-55. doi:10.4251/wjgo.v10.i1.48 Please consider including the last update about imatinib adjuvant therapy: Joensuu H, Eriksson M, Hall KS, et al. Three versus one year of adjuvant imatinib for high-risk gastrointestinal stromal tumor (GIST): Survival analysis of a randomized trial after 10 years of follow-up. Journal of Clinical Oncology. 2020; 38(15_suppl): 11503–11503, doi: 10.1200/jco.2020.38.15_suppl.11503. I would suggest making some small language corrections, mostly in terms of punctuation.
---

REVIEWER	Blay, Jean-Yves Centre Leon Berard, Department of Medical Oncology, Centre Léon Berard
REVIEW RETURNED	25-Apr-2021

GENERAL COMMENTS	This is an interesting project and an important question for these diseases. Several questions are still pending and a bit unclear yet, for the project to be successful. 1) the number of series is limited: why not envisage to collect the source data instead of publication data? 2) What is the primary surgical outcome, and what are the secondary endpoints? 3) Reference list requires updating, with more recent publications series and guidelines... 4) How will the author ensure to avoid multiple occurrence of a single patient in a institution in the present version of the project? 5) have the authors already figured out how many series/ patents they may have to collect?
---

VERSION 1 – AUTHOR RESPONSE

Answer to the reviewer:

Reviewer 1:

(1) Please consider including some more available papers dedicated to comparison of open and laparoscopic surgery in GIST treatment into references.

Answer: Sorry for our false that we did not provide the more available and latest references in our primary manuscript. Thus, we have modified the references in this version, which includes all you mentioned in previous suggestions. Thanks for your efforts to make our research better.

(2) I would suggest making some small language corrections, mostly in terms of punctuation.

Answer: We feel sorry that our inaccurate language in the manuscript maybe influences your reading experience. Our manuscript has been polished by a native speaker this time to make it more fluent and punctual. All the words or punctuations which were modified are marked in red in our newest version. Thanks again for your tolerance and kindness.

Reviewer 2:

(1) The number of series is limited: why not envisage to collect the source data instead of publication data?

Answer: Thanks for your rigorous consideration. Besides publication data, we will try to enlarge our including criteria to make more data included, such as conference, academic dissertation, and other grey literatures which are in process. We will contact study authors to obtain or confirm data to make the review more complete, aiming to enhance precision and reduce the impact of reporting biases. Thanks again for your scientific rigorousness.

(2) What is the primary surgical outcome, and what are the secondary endpoints?

Answer: We feel sorry for the inconvenience brought to the reviewer. Our network meta-analysis aims to compare the safety and efficacy of those procedures. The primary outcomes will be DFS, positive margin rate, and tumour rupture. Regarding surgical outcomes, the primary surgical outcomes are procedure time and surgical blood loss; the second surgical endpoint was whether or not there was conversion to another reaction. Thanks for your suggestions which make our study more clear and precise.

(3) Reference list requires updating, with more recent publications series and guidelines.

Answer: Thanks so much for your careful check. In the latest version, we have updated the more recent references. Any changes we have made have been marked in red.

(4) How will the author ensure to avoid multiple occurrence of a single patient in an institution in the present version of the project?

Answer: Thanks for your kind suggestions which will reduce the possibility of bias. In fact, such problem we have taken into consideration, but we are sorry we missed it in our precious manuscript. In our data extraction, we will record all the data sources which the including article provides. If there were articles from the same databases or identical hospitals, we would review the including criteria and study period to make sure whether a single patient was overlapped in such articles again. If not, we would include all. If yes, efforts will be made to contact the authors to get answers. Otherwise, only the latest or the most comprehensive one will be included. Thanks for your suggestions to improve the quality of our article.

(5) Have the authors already figured out how many series/ patents they may have to collect?

Answer: Thanks for your constructive suggestions. In our preliminary searching course, at least sixty-eight articles have met our including criteria, which including 8282 patients. Considering the rare prevalence of gastrointestinal stromal tumors, we think such considerable data is enough for network meta-analysis. We have collected the essential characteristics of these articles above. It will serve as a supplementary when we upload our newest manuscript for review. In addition, data extraction has not been started, for possible changes may be made in our manuscript according to reviewers' suggestions. Moreover, since the twice searching and additional searching in other sources is undergoing, the including articles which we have uploaded are not the final version in our network meta-analysis, which means we may add or omit some in these articles mentioned. We appreciate much for your suggestions.